# Monte Carlo Tree Search with Boltzmann Exploration

**Michael Painter, Mohamed Baioumy, Nick Hawes, Bruno Lacerda**
Oxford Robotics Institute
University of Oxford
{mpainter, mohamed, nickh, bruno}@robots.ox.ac.uk

## Abstract

Monte-Carlo Tree Search (MCTS) methods, such as Upper Confidence Bound applied to Trees (UCT), are instrumental to automated planning techniques. However, UCT can be slow to explore an optimal action when it initially appears inferior to other actions. Maximum ENtropy Tree-Search (MENTS) incorporates the maximum entropy principle into an MCTS approach, utilising *Boltzmann policies* to sample actions, naturally encouraging more exploration. In this paper, we highlight a major limitation of MENTS: optimal actions for the maximum entropy objective do not necessarily correspond to optimal actions for the original objective. We introduce two algorithms, Boltzmann Tree Search (BTS) and Decaying ENtropy Tree-Search (DENTS), that address these limitations and preserve the benefits of Boltzmann policies, such as allowing actions to be sampled faster by using the Alias method. Our empirical analysis shows that our algorithms show consistent high performance across several benchmark domains, including the game of Go.

## 1 Introduction

Planning under uncertainty is a core problem in Artificial Intelligence, commonly modelled as a Markov Decision Process (MDP) or variant thereof. MDPs can be solved using dynamic programming techniques to obtain an optimal policy [3]. However, computing a full optimal policy does not scale to large state-spaces, necessitating the use of heuristic solvers [18, 4] and online, sampling-based, planners based on Monte-Carlo Tree-Search (MCTS), such as the Upper Confidence Bound applied to Trees (UCT) algorithm [22].

The UCT search policy is designed to minimise *cumulative regret*, so manages a trade-off between exploration and exploitation. To exploit, UCT often selects the same action on successive trials, which can result in it getting stuck in local optima. Conversely, Maximum ENtropy Tree Search (MENTS) places a greater emphasis on exploration by combining MCTS with techniques from maximum entropy policy optimisation [38, 16, 17]. MENTS jointly maximises cumulative rewards and policy entropy, where a *temperature* parameter controls the weight of the entropy objective. However, MENTS is sensitive to this temperature parameter, and may not converge to the reward maximising policy or require a prohibitively low temperature to do so.

In this work, we consider scenarios where MCTS methods are used with a simulator to plan how an agent should act. We introduce two algorithms for this scenario that address the above limitations. First, we present Boltzmann Tree Search (BTS) which uses a Boltzmann search policy like MENTS, but optimises for reward maximisation only. Secondly, we introduce Decaying ENtropy Tree Search (DENTS), which adds entropy backups to BTS, but is still *consistent* (i.e. it converges to the reward maximising policy in the limit).

The main contributions of this paper are: (1) Demonstrating that the maximum entropy objective used in MENTS can be misaligned with reward maximisation, thus preventing it from converging to the optimal policy; (2) Introducing two new algorithms, BTS and DENTS, which preserve the benefits

37th Conference on Neural Information Processing Systems (NeurIPS 2023).

of using Boltzmann search policies while being as simple to implement as UCT and MENTS, but converge to the reward maximising policy; (3) Analysing MENTS, BTS and DENTS through the lens of *simple regret* to provide theoretical convergence results; (4) Highlighting and demonstrating that the Alias method [35, 34] can be used with stochastic action selection to improve the asymptotic complexity of running a fixed number of trials over existing MCTS algorithms; and (5) Demonstrating the performance improvements of Boltzmann search policies used in BTS and DENTS in benchmark gridworld environments and the game of Go.

## 2 Background

### 2.1 Markov Decision Processes

We define a (finite-horizon) MDP as a tuple $\mathcal{M} = (\mathcal{S}, \mathcal{A}, p, R, H)$, where $\mathcal{S}$ is the set of states; $\mathcal{A}$ is the set of actions; $p : \mathcal{S} \times \mathcal{A} \times \mathcal{S} \to [0, 1]$ is the transition function where $p(s'|s, a, )$ is the probability of moving to state $s'$ given that action $a$ was taken in state $s$; $R : \mathcal{S} \times \mathcal{A} \to \mathbb{R}$ is the reward function; and $H \in \mathbb{N}$ is the finite horizon. Let $\mathrm{Succ}(s, a)$ denote the set of successor states of $(s, a)$, i.e. $\mathrm{Succ}(s, a) = \{s' \in \mathcal{S} \mid p(s'|s, a) > 0\}$.

A policy $\pi$ maps a state and timestep to a distribution over actions, and we denote the probability of executing $a$ at state $s$ and timestep $t$ as $\pi(a|s, t)$. Let $s_t$ denote the state after $t$ time-steps and $a_t$ the action selected at $s_t$, according to $\pi$. The expected value $V^\pi$ and expected state-action value $Q^\pi$ of $\pi$ are defined as:

$$V^\pi(s, t) = \mathbb{E}_\pi \left[ \sum_{i=t}^{H} R(s_i, a_i) \Big| s_t = s \right], \tag{1}$$

$$Q^\pi(s, a, t) = R(s, a) + \mathbb{E}_{s' \sim p(\cdot|s, a)}[V^\pi(s', t+1)]. \tag{2}$$

The goal is to find the *optimal policy* $\pi^*$ with the maximum expected reward: $\pi^* = \arg\max_\pi V^\pi$. The optimal value functions are then defined as $V^* = V^{\pi^*}, Q^* = Q^{\pi^*}$. For an MDP, there always exists an optimal policy $\pi^*$ which is deterministic [24].

### 2.2 Maximum entropy policy optimization

In planning and reinforcement learning, the agent usually aims to maximise the expected sum of rewards. In maximum entropy policy optimisation, the objective is augmented with the expected entropy of the policy [16, 38]. Formally, this is expressed as:

$$V_{\mathrm{sft}}^\pi(s, t) = \mathbb{E}_\pi \left[ \sum_{i=t}^{H} R(s_i, a_i) + \alpha \mathcal{H}(\pi(\cdot|s_i, i)) \Big| s_t = s \right], \tag{3}$$

where $\alpha \geq 0$ is a temperature parameter, and $\mathcal{H}$ is the Shannon entropy function. The temperature determines the relative importance of the entropy against the reward and thus controls the stochasticity of the optimal policy. The conventional reward maximisation objective can be recovered by setting $\alpha = 0$.

An optimal value function for maximum entropy optimization is obtained using the *soft* Bellman optimality equations [17]:

$$Q_{\mathrm{sft}}^*(s, a, t) = R(s, a) + \mathbb{E}_{s' \sim p(\cdot|s, a)}\left[V_{\mathrm{sft}}^*(s', t)\right], \tag{4}$$

$$V_{\mathrm{sft}}^*(s, t) = \alpha \log \sum_{a \in \mathcal{A}} \exp(Q_{\mathrm{sft}}^*(s, a, t)/\alpha), \tag{5}$$

which corresponds to a standard Bellman backup, with the $\max$ replaced by a *softmax*, shown in Equation (5). The optimal soft policy $\pi_{\mathrm{sft}}^* = \arg\max_\pi V_{\mathrm{sft}}^\pi$ can be computed directly [26] as follows:

$$\pi_{\mathrm{sft}}^*(a|s, t) = \exp((Q_{\mathrm{sft}}^*(s, a, t) - V_{\mathrm{sft}}^*(s, t))/\alpha). \tag{6}$$

Note that the soft policy is always stochastic for any $\alpha > 0$. Henceforth, we will use *soft value* to refer to value functions indexed with 'sft', *(optimal) soft policy* to refer to policies of the form given in Equation (6), and *standard value* and *(optimal) standard policy* for values and policies of the form given in Section 2.1, unless it is clear from the context.

For the remainder of this paper we will drop the timestep $t$ from policies and value functions to simplify notation.

## 2.3  Monte-Carlo tree search

MCTS methods build a search tree $\mathcal{T}$ using Monte-Carlo trials. Each trial is split into two phases: starting from the root node, actions are chosen according to a *search policy* and states sampled from the transition distribution until the first state not in $\mathcal{T}$ is reached. A new node is added to $\mathcal{T}$ and its value is initialised using some function $V^{\mathrm{init}}$, often using a *rollout policy* to select actions until the time horizon $H$ is reached. In the second phase, the return for the trial is back-propagated up (or 'backed up') the tree to update the values of nodes in $\mathcal{T}$. For a reader unfamiliar with MCTS, we refer to [6] for a review of the MCTS literature, as many variants of MCTS exist and may vary from our description.

Two critical choices in designing an MCTS algorithm are the search policy (which needs to balance exploration and exploitation) and the backups (how values are updated). MCTS algorithms are often designed to achieve *consistency* (i.e. convergence to the optimal action in the limit), which implies that running more trials will increase the probability that the optimal action is recommended.

To simplify notation we assume that each node in the search tree corresponds to a unique state, so we may represent nodes using states. Our algorithms and results do not make use of this assumption, and generalise to when this assumption does not hold.

**UCT**  UCT [22] applies the upper confidence bound (UCB) in its search policy to balance exploration and exploitation. The $n$th trial of UCT operates as follows: let $\mathcal{T}$ be the current search tree and let $\tau = (s_0, a_0, ..., a_{h-1}, s_h)$ denote the trajectory of the $n$th trial, where $s_h \notin \mathcal{T}$ or $h = H$. At each node $s_t$ the UCT search policy $\pi_{\mathrm{UCT}}$ will select a random action that has not previously been selected, otherwise, it will select the action with maximum UCB value:

$$\pi_{\mathrm{UCT}}(s) = \max_{a \in \mathcal{A}} \bar{Q}(s, a) + c\sqrt{\frac{\log N(s)}{N(s, a)}}, \tag{7}$$

where, $\bar{Q}(s, a)$ is the current empirical Q-value estimate, $N(s)$ (and $N(s, a)$) is how many times $s$ has been visited (and action $a$ selected) and $c$ is an exploration parameter. Then, $s_h$ is added to the tree: $\mathcal{T} \leftarrow \{s_h\} \cup \mathcal{T}$. The backup consists of updating empirical estimates for $t = h - 1, ..., 0$:

$$\bar{Q}(s_t, a_t) \leftarrow \bar{Q}(s_t, a_t) + \frac{\bar{R}(t) - \bar{Q}(s_t, a_t)}{N(s_t, a_t) + 1}, \tag{8}$$

where $\bar{R}(t) = V^{\mathrm{init}}(s_h) + \sum_{i=t}^{h-1} R(s_i, a_i)$, and $V^{\mathrm{init}}(s_h) = \sum_{i=h}^{H} R(s_i, a_i)$ if using a rollout policy.

**MENTS**  MENTS [37] combines maximum entropy policy optimization [16, 38] with MCTS. Algorithmically, it is similar to UCT. The two differences are: (1) the search policy follows a stochastic Boltzmann policy, and (2) it uses soft values that are updated with dynamic programming backups. The MENTS search policy $\pi_{\mathrm{MENTS}}$ is given by:

$$\pi_{\mathrm{MENTS}}(a|s) = (1 - \lambda_s)\rho_{\mathrm{MENTS}}(a|s) + \frac{\lambda_s}{|\mathcal{A}|}, \tag{9}$$

$$\rho_{\mathrm{MENTS}}(a|s) = \exp\left(\frac{1}{\alpha}\left(\hat{Q}_{\mathrm{sft}}(s, a) - \hat{V}_{\mathrm{sft}}(s)\right)\right) \tag{10}$$

where $\lambda_s = \min(1, \epsilon/\log(e + N(s)))$, $\epsilon \in (0, \infty)$ is an exploration parameter and $\hat{V}_{\mathrm{sft}}(s)$ (and $\hat{Q}_{\mathrm{sft}}(s, a)$) are the current soft (Q-)value estimates. The soft value of the new node is initialised $\hat{V}_{\mathrm{sft}}(s_h) \leftarrow V^{\mathrm{init}}(s_h)$ and the soft values are updated with backups for $t = h - 1, ..., 0$:

$$\hat{Q}_{\mathrm{sft}}(s_t, a_t) \leftarrow R(s_t, a_t) + \sum_{s' \in \mathrm{Succ}(s,a)} \left(\frac{N(s')}{N(s_t, a_t)}\hat{V}_{\mathrm{sft}}(s')\right), \tag{11}$$

$$\hat{V}_{\mathrm{sft}}(s_t) \leftarrow \alpha \log \sum_{a \in \mathcal{A}} \exp\left(\frac{1}{\alpha}\hat{Q}_{\mathrm{sft}}(s_t, a)\right). \tag{12}$$

Each $\hat{Q}_{\mathrm{sft}}(s, a)$ is initialised using another function $Q_{\mathrm{sft}}^{\mathrm{init}}(s, a)$ (but is typically zero).

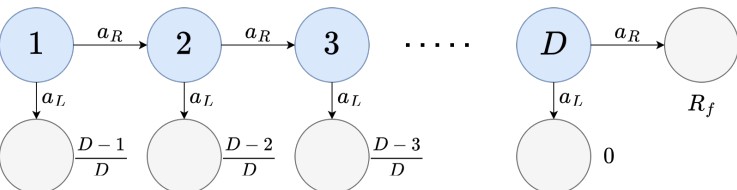

Figure 1: An illustration of the *(modified) D-chain problem*, where 1 is the starting state, transitions are deterministic and values next to states represent rewards for arriving in that state.

## 2.4 Simple regret

UCB [1] is frequently used in MCTS methods to minimise *cumulative regret* during the tree search. Cumulative regret is most appropriate in scenarios where the actions taken during tree search have an associated real-world cost. However, MCTS methods often use a simulator during the tree search, where the only significant real-world cost is associated with taking the recommended action after the tree search. In such scenarios, simple regret [7, 8] is more appropriate for analysing the performance of algorithms, as it only considers the cost of the actions that are actually executed. Under simple regret, algorithms are not penalised for under-exploiting during the search, thus can explore more, which leads to better recommendations by allowing algorithms to confirm that bad actions are indeed of lower value.

We consider the problem of MDP planning as a sequential decision problem, where for each round $n$:

1. the forecaster algorithm produces a search policy $\pi^n$ and samples a trajectory $\tau \sim \pi^n$,
2. the environment returns the rewards $R(s_t, a_t)$ for each $s_t, a_t$ pair in $\tau$,
3. the forecaster algorithm produces a recommendation policy $\psi^n$,
4. if environment sends stop signal, then end, else return to step 1.

The *simple regret* of the forecaster on it's $n$th round is then:

$$\text{reg}(s, \psi^n) = V^*(s) - V^{\psi^n}(s). \tag{13}$$

In MENTS the recommendation policy suggested in [37] can be written:

$$\psi_{\text{MENTS}}(s) = \arg\max_{a \in \mathcal{A}} \hat{Q}_{\text{sft}}(s, a). \tag{14}$$

We can now formally define *consistency*: an algorithm is *consistent* if and only if its recommendation policy $\psi^n$ converges to an expected simple regret of zero: $\mathbb{E}[\text{reg}(s, \psi^n)] \to 0$ as $n \to \infty$. Note that because we are considering randomised algorithms, there is a distribution over the possible recommendation policies that could have been produced. If a policy has a simple regret of zero then it implies it is an optimal policy.

## 3 Limitations of prior MCTS methods

In this section we use the *D-chain problem* introduced in [9] (Figure 1) to highlight the limitations of UCT and MENTS. In the D-chain problem, when an agent chooses action $a_L$, from some state $d$, it moves to an absorbing state and receives a reward of $(D - d)/D$. In state $D$, action $a_R$ corresponds to an absorbing state with reward $R_f = 1$. The optimal standard policy always selects action $a_R$.

In the 10-chain problem ($D = 10$), UCT will recommend action $a_L$ from state 1 (Figure 2a). UCT requires $\Omega(\exp(...\exp(1)...))$ many trials ($D$ composed exponential functions) to recommend the optimal policy that reaches the reward of $R_f = 1$ [9]. This highlights the first limitation mentioned in Section 1: UCT quickly disregards action $a_R$ at the initial state, to exploit the reward of 0.9.

When MENTS is run on the 10-chain problem, with the help of the entropy term it quickly finds the final reward of $R_f = 1$ (Figure 2a). However, consider the *modified 10-chain* with $R_f = 1/2$ instead. Repeated applications of Equations (4) and (5) for $\alpha = 1$ gives the optimal soft values of $Q^*_{\text{sft}}(1, a_R) = \log(\exp(1/2) + \sum_{i=0}^{8} \exp(i/10)) \approx 2.74$ and $Q^*_{\text{sft}}(1, a_L) = 0.9$. So in the modified

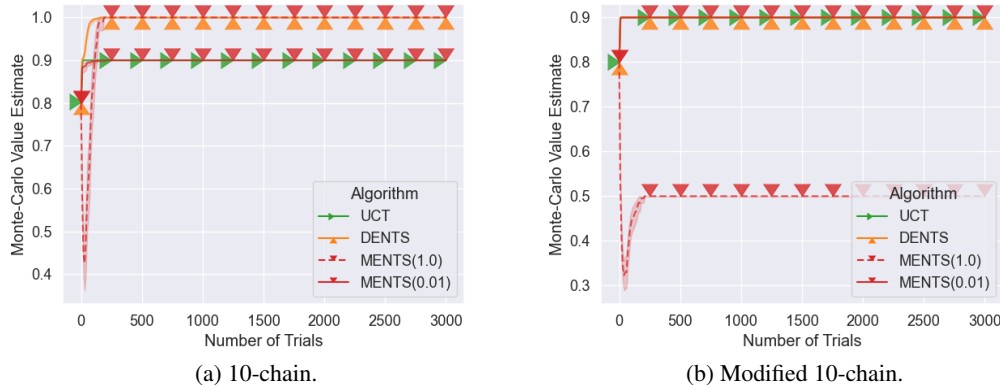

|   (a) 10-chain.   |   (b) Modified 10-chain.   |

Figure 2: A comparison of MENTS, DENTS and UCT when run on the (modified) 10-chain.

10-chain problem, we have $Q_{\text{sft}}^*(1, a_R) > Q_{\text{sft}}^*(1, a_L)$ with $\alpha = 1$, whereas $Q^*(1, a_R) < Q^*(1, a_L)$. Thus, when MENTS converges, it will recommend the wrong action with respect to the standard objective (Figure 2b), i.e. it is not consistent. The modified 10-chain is an example of Proposition 3.1, which states that MENTS will not always converge to the standard optimal policy.

**Proposition 3.1.** *There exists an MDP $\mathcal{M}$ and temperature $\alpha$ such that $\mathbb{E}[\text{reg}(s_0, \psi_{\text{MENTS}}^n)] \nrightarrow 0$ as $n \to \infty$. That is, MENTS is not consistent.*

*Proof.* Proof is by example with $\alpha = 1$ in the modified 10-chain (Figure 1). □

We can reduce the value of $\alpha$ to decrease the importance of entropy in the soft objective $\mathbb{E}[R(s_t, a_t) + \alpha \mathcal{H}(\pi(\cdot|s_t))]$. If $\alpha$ is small enough, then MENTS recommendations can converge to the optimal standard policy (Theorem 3.2). Hence, MENTS with a low temperature can solve the modified 10-chain problem (Figure 2b). However, in practice, a low temperature will often cause MENTS to not sufficiently explore, as demonstrated in the original D-chain (Figure 2a).

**Theorem 3.2.** *For any MDP $\mathcal{M}$, after running $n$ trials of the MENTS algorithm with $\alpha \leq \Delta_{\mathcal{M}}/3H \log |\mathcal{A}|$, there exists constants $C, k > 0$ such that: $\mathbb{E}[\text{reg}(s_0, \psi_{\text{MENTS}})] \leq C \exp(-kn)$, where $\Delta_{\mathcal{M}} = \min\{Q^*(s, a) - Q^*(s, a')|Q^*(s, a) \neq Q^*(s, a'), s \in \mathcal{S}, a, a' \in \mathcal{A}, t \in \mathbb{N}\}$.*

*Proof outline.* We can show $\hat{Q}_{\text{sft}}(s, a) \xrightarrow{p} Q_{\text{sft}}^*(s, a)$ (Corollary E.12.1) similarly to Thoerem 4.2. The bound on $\alpha$ is required to ensure that $\pi^*(s_0) = \pi_{\text{sft}}^*(s_0)$. □

In conclusion, similar MDPs can require vastly different temperatures for MENTS to be effective. We discuss MENTS sensitivity to the temperature parameter further in Appendix D.2, and demonstrate this parameter sensitivity in the Frozen Lake environment (Section 5.1) in Figure 27 in the appendix.

## 4 Boltzmann search

We now introduce two algorithms that utilise Boltzmann search policies similar to MENTS and admit bounded simple regrets that converge to zero without restrictive constraints on parameters. Thus, they do not suffer from sensitivity to parameter selection that MENTS does. Both algorithms use action selection and value backups that are easy to implement and use. We designed these algorithms with consistency in mind, which in practice, means that if we run more trials then we (with high probability) will recommend a better solution (note that Proposition 3.1 implies that this is not always the case for MENTS).

### 4.1 Boltzmann Tree Search

Our first approach, put simply, replaces the use of soft values in MENTS with *Bellman* values. We call this algorithm *Boltzmann Tree Search* (BTS). BTS promotes exploration through the stochastic

Boltzmann search policy, like MENTS, while using backups that optimise for the standard objective, like UCT. The search policy $\pi_{\text{BTS}}$ and backups for the $n$th trial are given by:

$$\pi_{\text{BTS}}(a|s) = (1 - \lambda_s)\rho_{\text{BTS}}(a|s) + \frac{\lambda_s}{|\mathcal{A}|}, \tag{15}$$

$$\rho_{\text{BTS}}(a|s) \propto \exp\left(\frac{1}{\alpha}\left(\hat{Q}(s,a)\right)\right), \tag{16}$$

$$\hat{Q}(s_t, a_t) \leftarrow R(s_t, a_t) + \sum_{s' \in \text{Succ}(s_t, a_t)}\left(\frac{N(s')}{N(s_t, a_t)}\hat{V}(s')\right), \tag{17}$$

$$\hat{V}(s_t) \leftarrow \max_{a \in \mathcal{A}}\hat{Q}(s_t, a), \tag{18}$$

for $t = h - 1, ..., 0$, where $\hat{V}$ and $\hat{Q}$ are the current Bellman (Q-)value estimates, $\lambda_s = \min(1, \epsilon/\log(e + N(s)))$, $\epsilon \in (0, \infty)$ is an exploration parameter and $\alpha$ is a search temperature (unrelated to entropy). Each $\hat{V}(s)$ and $\hat{Q}(s,a)$ are initialised using $V^{\text{init}}$ and $Q^{\text{init}}$ functions similarly to MENTS. The Bellman values are used for recommendations:

$$\psi_{\text{BTS}}(s) = \arg\max_{a \in \mathcal{A}}\hat{Q}(s, a). \tag{19}$$

By using Bellman backups, we can guarantee that the BTS recommendation policy converges to the optimal standard policy for any temperature $\alpha$, given enough time. In other words, BTS is consistent.

**Theorem 4.1.** *For any MDP $\mathcal{M}$, after running $n$ trials of the BTS algorithm with a root node of $s_0$, there exists constants $C, k > 0$ such that for all $\varepsilon > 0$ we have $\mathbb{E}[\text{reg}(s_0, \psi_{\text{BTS}})] \leq C \exp(-kn)$, and also $\hat{V}(s_0) \xrightarrow{p} V^*(s_0)$ as $n \to \infty$.*

*Proof outline.* This result is a special case of Theorem 4.2 by setting $\beta(m) = 0$. ☐

## 4.2 Decaying Entropy Tree Search

Secondly, we present Decaying ENtropy Tree Search (DENTS), which can effectively interpolate between the MENTS and BTS algorithms. DENTS also uses the dynamic programming backups from equations (17) and (18), but adds an *entropy backup*. The *entropy values* are weighted by a bounded non-negative function $\beta(N(s))$ in the DENTS search policy $\pi_{\text{DENTS}}$:

$$\pi_{\text{DENTS}}(a|s) = (1 - \lambda_s)\rho_{\text{DENTS}}(a|s) + \frac{\lambda_s}{|\mathcal{A}|}, \tag{20}$$

$$\rho_{\text{DENTS}}(a|s) \propto \exp\left(\frac{1}{\alpha}\left(\hat{Q}(s,a) + \beta(N(s))\mathcal{H}_Q(s,a)\right)\right), \tag{21}$$

$$\mathcal{H}_V(s_t) \leftarrow \mathcal{H}(\pi_{\text{DENTS}}(\cdot|s_t)) + \sum_{a \in \mathcal{A}}\pi_{\text{DENTS}}(a|s_t)\mathcal{H}_Q(s_t, a), \tag{22}$$

$$\mathcal{H}_Q(s_t, a_t) \leftarrow \sum_{s' \in \text{Succ}(s_t, a_t)}\frac{N(s')}{N(s_t, a_t)}\mathcal{H}_V(s'), \tag{23}$$

for $t = h - 1, ..., 0$, where $\mathcal{H}_V(s)$ and $\mathcal{H}_Q(s, a)$ are the *entropy values* of the search policy rooted at $s$ and $(s, a)$ respectively, and $\alpha, \lambda_s$ are the same as for BTS, as described in Section 4.1. Initial values are the same as Section 4.1, and the entropy values are initialised to zero. In DENTS we can view $\hat{Q}(s, a) + \beta(N(s))\mathcal{H}_Q(s, a)$ as a soft value for $(s, a)$. Hence, by setting $\beta(m) = \alpha$, the DENTS search will mimic the MENTS search (demonstrated in Appendix D.4), and if $\beta(m) = 0$ then the algorithm reduces to the BTS algorithm. By using a decaying function for $\beta$ we amplify values using entropy as an exploration bonus early in the search while allowing for more exploitation later. Recommendations still use Bellman values:

$$\psi_{\text{DENTS}}(s) = \arg\max_{a \in \mathcal{A}}\hat{Q}(s, a). \tag{24}$$

Because the recommendation policy $\psi_{\text{DENTS}}$ uses the Bellman values, we can guarantee that it will converge to the optimal standard policy, and is consistent for any $\beta$.

**Theorem 4.2.** *For any MDP $\mathcal{M}$, after running $n$ trials of the DENTS algorithm with a root node of $s_0$, if $\beta$ is a bounded function, then there exists constants $C, k > 0$ such that for all $\varepsilon > 0$ we have $\mathbb{E}[\mathrm{reg}(s_0, \psi_{\mathrm{DENTS}})] \leq C \exp(-kn)$, and also $\hat{V}(s_0) \xrightarrow{p} V^*(s_0)$ as $n \to \infty$.*

*Proof outline.* Let $0 < \eta < \pi_{\mathrm{DENTS}}(a|s)$ for all $s, a$, which exists because $\exp(\cdot) > 0$ and $1/|\mathcal{A}| > 0$ in Equation 20 (Lemma E.1), which can be used with Hoeffding bounds to show for appropriate constants and any event $E$ that $\{\Pr(E) \leq C_0 \exp(-k_0 \varepsilon^2 N(s_t))\}$ iff $\{\Pr(E) \leq C_1 \exp(-k_1 \varepsilon^2 N(s_t, a_t))\}$ iff $\{\Pr(E) \leq C_2 \exp(-k_2 \varepsilon^2 N(s_{t+1}))\}$. We can then show $\Pr(|\hat{Q}(s_0, a) - Q^*(s_0, a)| > \varepsilon) < C_3 \exp\left(-k_3 \varepsilon^2 n\right)$ by induction, where the base case $\hat{V}(s_{H+1}) = V^*(s_{H+1}) = 0$ holds vacuously (Lemmas E.10, E.16 and Theorem E.17). Let $\Delta_{\mathcal{M}}$ be a small constant (Equation (122)) such that $\forall s, a. |\hat{Q}(s_0, a) - Q^*(s_0, a)| \leq \Delta_{\mathcal{M}}/2 \implies \arg\max_a \hat{Q}(s, a) = \arg\max_a Q^*(s, a)$. Setting $\varepsilon = \Delta_{\mathcal{M}}/2$ gives a bound on $\Pr(\psi_{\mathrm{DENTS}} \neq \pi^*)$, which can then be used in the definition of simple regret to give the result. $\square$

### 4.3 Using the Alias method

The Alias method [35, 34] can be used to sample from a categorical distribution with $m$ categories in $O(1)$ time, with a preprocessing step of $O(m)$ time. Given any stochastic search policy, we can sample actions in amortised $O(1)$ time, by computing an alias table every $|\mathcal{A}|$ visits to a node, and then sampling from that table. Note that when using the Alias method we are making a trade off between using the most up to date policy and the speed of sampling actions.

In Appendix C.1 we discuss this idea in more detail, and give an informal analysis of the complexity to run $n$ trials. BTS, DENTS and MENTS can run $n$ trials in $O(n(H \log(|\mathcal{A}|) + |\mathcal{A}|))$ time when using the Alias method, as opposed to the typical complexity of $O(nH|\mathcal{A}|)$.

### 4.4 Limitations and benefits

The main limitations of BTS and DENTS are as follows: (1) the DENTS decay function $\beta$ can be non-trivial to set and tune; (2) the focus on simple regret and exploration means they are not appropriate to use when actions taken during the tree search/planning phase have a real-world cost; (3) the backups implemented directly as presented above are computationally more costly than computing the average returns that UCT uses; (4) when it is desirable for an agent to follow the maximum entropy policy, then MENTS would be preferable, for example if the agent needs to explore to learn and discover an unknown environment.

The main benefits of using a stochastic policy for action selection are: (1) they allow the Alias method (Section 4.3) to be used to speed up trials; (2) they naturally encourage exploration as actions are sampled randomly, which is useful for discovering sparse or delayed rewards and for confirming that actions with low values do in fact have a low value; and (3) the entropy of a stochastic policy can be computed and used as an exploration bonus. In Appendix C.3 we summarise and compare the differences between the algorithms considered in this work in more detail.

## 5 Results

This section compares the proposed BTS and DENTS against MENTS and UCT on a set of goal-based MDPs and in the game of Go. For additional baselines, we also compare with the RENTS and TENTS algorithms [10], which use *relative* and *Tsalis* entropy in place of Shannon entropy respectively, and the H-MCTS algorithm [20] which combines UCT and Sequential Halving.

### 5.1 Gridworlds

To evaluate an algorithm with search tree $\mathcal{T}$, we complete the partial recommendation policy $\psi_{\mathrm{alg}}$ as follows:

$$\psi(a|s) = \begin{cases} 1 & \text{if } s \in \mathcal{T} \text{ and } a = \psi_{\mathrm{alg}}(s), \\ 0 & \text{if } s \in \mathcal{T} \text{ and } a \neq \psi_{\mathrm{alg}}(s), \\ \frac{1}{|\mathcal{A}|} & \text{otherwise.} \end{cases} \tag{25}$$

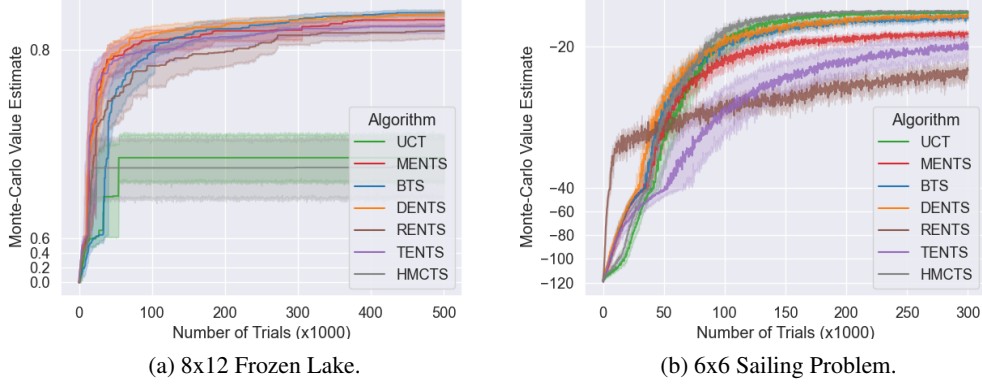

(a) 8x12 Frozen Lake.           (b) 6x6 Sailing Problem.

Figure 3: Results for gridworld environments. Further results are given in Appendix D.3.

We sample a number of trajectories from $\psi$, and take the average return to estimate $V^\psi$. Although we are evaluating the algorithms in an *offline planning* setting, it still indicates how the algorithms perform in an *online* setting where we interleave planning in simulation with letting the agent act.

### 5.1.1 Domains

To validate our approach, we use the Frozen Lake environment [5], and the Sailing Problem [28], commonly used to evaluate tree search algorithms [28, 22, 33, 13]. We chose these environments to compare our algorithms in a domain with a sparse and dense reward respectively.

The *(Deterministic) Frozen Lake* is a grid world environment with one goal state. The agent can move in any cardinal direction at each time step, and walking into a wall leaves the agent in the same location. Trap states exist where the agent falls into a hole and the trial ends. If the agent arrives at the goal state after $t$ timesteps, then a reward of $0.99^t$ is received.

The *Sailing Problem* is a grid world environment with one goal state, at the opposite corner to the starting location of the agent. The agent has 8 different actions to travel each of the 8 adjacent states. In each state, the wind is blowing in a given direction and will stochastically change after every transition. The agent cannot sail directly into the wind. The cost of each action depends on the *tack*, the angle between the direction of the agent's travel and the wind.

### 5.1.2 Results

We used an 8x12 Frozen Lake environment and a 6x6 Sailing Problem for evaluation, more environment details are given in Appendix D.1. Parameters were selected using a hyper-parameter search (Appendix D.3). Each algorithm is run 25 times on each environment and evaluated every 250 trials using 250 trajectories. A horizon of 100 was used for Frozen Lake and 50 for the Sailing Problem.

In Frozen Lake (Figure 3a), entropy proved to be a useful exploration bonus for the *sparse reward*. Values in UCT and BTS remain at zero until a trial successfully reaches the goal. However, entropy guides agents to avoid trap states, where the entropy is zero. DENTS was able to perform similarly to MENTS, and BTS was able to improve its policy over time more than UCT.

In the Sailing Problem (Figure 3b) UCT performs well due to the dense reward. BTS and DENTS also manage to keep up with UCT. MENTS and TENTS appear to be slightly hindered by entropy in this environment. The relative entropy encourages RENTS to pick the same actions over time, so it tends to pick a direction and stick with it regardless of cost.

Finally, BTS and DENTS were able to perform well in both domains with a sparse and dense reward structure, whereas the existing methods performed better on one than the other, hence making BTS and DENTS good candidates for a general purpose MCTS algorithm.

## 5.2 Go

For a more challenging domain we ran a round-robin tournament using the game of Go, which has widely motivated the development of MCTS methods [14, 30, 32]. In each match, each algorithm played 50 games as black and 50 as white. *Area scoring* is used to score the games, with a *komi* of 7.5. We used an openly available value network $\tilde{V}$ and policy network $\tilde{\pi}$ from KataGo [36]. Our baseline was the PUCT algorithm [2], as described in Alpha Go Zero [32] using prioritised UCB [29] to utilise the policy neural network. Each algorithm was limited to 5 seconds of compute time per move, allowed to use 32 search threads per move, and had access to 80 Intel Xeon E5-2698V4 CPUs clocked at 2.2GHz, and a single Nvidia V100 GPU on a shared compute cluster.

To use Boltzmann search in Go, we adapted the algorithms to account for an opponent that wishes to minimise the value of a two-player game. This is achieved by appropriately negating values used in the search policy and backups, which is described precisely in Appendix C.2.

Additionally, we found that adapting the algorithms to use average returns (recall Equation (8)) outperformed using Bellman backups for Go (Appendix D.5.1). The Bellman backups were sensitive to and propogated noise from the neural network evaluations. We use the prefix 'AR' to denote the algorithms using average returns, such as AR-DENTS. Full details for these algorithms are given in Appendix B.

### 5.2.1 Using neural networks with Boltzmann search

This section describes how to use value and policy networks in BTS. Adapting MENTS and DENTS are similar (Appendix C.2). Values can be initialised with the neural networks as $\hat{Q}(s,a) \leftarrow \log \tilde{\pi}(a|s) + B$ and $\hat{V}(s) \leftarrow \tilde{V}(s)$, where $B$ is a constant (adapted from Xiao et al. [37]). With such an initialisation, the initial BTS policy is $\rho_{\text{BTS}}(a|s) \propto \tilde{\pi}(a|s)^{1/\alpha}$. For these experiments we set a value of $B = \frac{-1}{|\mathcal{A}|} \sum_{a \in \mathcal{A}} \log \tilde{\pi}(a|s)$. Additionally, the stochastic search policy naturally lends itself to mixing in a prior policy, so we can replace BTS search policy $\pi_{\text{BTS}}$ (Equation (15)) with $\pi_{\text{BTS,mix}}$:

$$\pi_{\text{BTS,mix}}(a|s) = \lambda_{\tilde{\pi}} \tilde{\pi}(a|s) + (1 - \lambda_{\tilde{\pi}}) \pi_{\text{BTS}}(a|s) \tag{26}$$

$$= \lambda_{\tilde{\pi}} \tilde{\pi}(a|s) + (1 - \lambda_{\tilde{\pi}})(1 - \lambda_s) \rho_{\text{BTS}}(a|s) + \frac{(1 - \lambda_{\tilde{\pi}})\lambda_s}{|\mathcal{A}|}, \tag{27}$$

where $\lambda_{\tilde{\pi}} = \min(1, \epsilon_{\tilde{\pi}} / \log(e + N(s)))$, and $\epsilon_{\tilde{\pi}} \in (0, \infty)$ controls the weighting for the prior policy.

### 5.2.2 Results

Results of the round-robin are summarised in Table 1, and we discuss how parameters were selected in Appendix D.5.1. BTS was able to run the most trials per move and beat all of the other algorithms other than DENTS which it drew. We used the optimisations outlined in Appendix C.1 which allowed the Boltzmann search algorithms to run significantly more trials per move than PUCT. BTS and DENTS were able to beat PUCT with results of 57-43 and 58-42 respectively. Using entropy did not seem to have much benefit in these experiments, as can be witnessed by MENTS only beating TENTS, and DENTS drawing 50-50 with BTS. This is likely because the additional exploration provided by entropy is vastly outweighed by utilising the information contained in the neural networks $\tilde{V}$ and $\tilde{\pi}$. Interestingly RENTS had the best performance out of the prior works, losing 43-57 to PUCT, and the use of relative entropy appears to take advantage of a heuristic for Go that the RAVE [15] algorithm used: the value of a move is typically unaffected by other moves on the board.

To validate the strength of our PUCT agent, we also compared it directly with KataGo [36], limiting each algorithm to 1600 trials per move. Our PUCT agent won 61-39 in 9x9 Go, and lost 35-65 in 19x19 Go, suggesting that our PUCT agent is strong enough to provide a meaningful comparison for our other general purpose algorithms. Finally, note that we did not fine-tune the neural networks, so the Boltzmann search algorithms directly used the networks that were trained for use in PUCT.

## 6 Related work

UCT [22, 23] is a widely used variant of MCTS. Polynomial UCT [2], replaces the logarithmic term in UCB with a polynomial one (such as a square root), which has been further popularised by its use

| Black \White | PUCT | AR-M | AR-R | AR-T | AR-B | AR-D | Trials/move |
|---|---|---|---|---|---|---|---|
| PUCT | - | 33-17 | 27-23 | 42-8 | 17-33 | 15-35 | 1054 |
| AR-MENTS | 12-48 | - | 13-37 | 38-12 | 10-40 | 12-38 | 4851 |
| AR-RENTS | 20-30 | 24-26 | - | 39-11 | 18-32 | 14-36 | 3672 |
| AR-TENTS | 8-42 | 11-39 | 9-41 | - | 6-44 | 10-40 | 5206 |
| AR-BTS | 25-25 | 35-15 | 31-19 | 34-16 | - | 15-35 | 5375 |
| AR-DENTS | 23-27 | 36-14 | 29-21 | 36-14 | 15-35 | - | 4677 |

Table 1: Results for the Go round-robin tournament. The first column gives the agent playing as black. The final column gives the average trials run per move across the entire round-robin. In the top row, we abbreviate the algorithm names for space.

in Alpha Go and Alpha Zero [30, 32, 31]. Coquelin and Munos introduce the Flat-UCB and BAST algorithms to adapt UCT for the D-chain problem [9]. However, we consider an alternative approach for search in MCTS rather than adapting UCB.

Maximum entropy policy optimization methods are well-known in the reinforcement learning literature [16, 17, 38]. MENTS [37] is the first method to combine the principle of maximum entropy and MCTS. Kozakowski et al. [25] extend MENTS to arrive at Adaptive Entropy Tree Search (ANTS), adapting parameters throughout the search to match a prescribed entropy value. Dam et al. [10] also extend MENTS using *Relative* and *Tsallis entropy* to arrive at the RENTS and TENTS algorithms. Our work is closely related to MENTS, however, we focus on reward maximisation and consider how entropy can be used in MCTS without altering our planning objective.

Bubeck et al. [7, 8] introduce simple regret in the context of multi-armed bandit problems (MABs). They alternate between pulling arms for exploration and outputting a *recommendation*. They show for MABs that a uniform exploration produces an exponential bound on the simple regret of recommendations. We use simple regret, but in the context of sequential decision-making, to analyse the convergence of MCTS algorithms.

Tolpin and Shimony [33] extend simple regret to MDP settings, showing an $O(\exp(-\sqrt{n}))$ bound on simple regret after $n$ trials by adapting UCT. The subsequent work of Hay et al. [19] extends [33] to consider a *metalevel decision problem*, incorporating computation costs into the objective. Pepels et al. [27] introduce a Hybrid MCTS (H-MCTS) motivated by the notion of simple regret. H-MCTS uses a mixture of Sequential Halving [20], and UCT. Feldman et al. [13, 11, 12] introduce the Best Recommendation with Uniform Exploration (BRUE) algorithm. BRUE splits trials up to explicitly focus on exploration and value estimation one at a time. BRUE achieves an exponential bound $O(\exp(-n))$ on simple regret after $n$ trials [13]. Prior work that considers simple regret in MCTS has focused on adaptations to UCT, whereas this work focuses on algorithms that sample actions from Boltzmann distributions, rather than using UCB for action seclection.

## 7   Conclusion

We considered the recently introduced MENTS algorithm, compared and contrasted it to UCT, and discussed the limitations of both. We introduced two new algorithms, BTS and DENTS, that are consistent, converge to the optimal standard policy, while preserving the benefits that come with using a stochastic Boltzmann search policy. Finally, we compared our algorithms in gridworld environments and Go, demonstrating the performance benefits of utilising the Alias method, that entropy can be a useful exploration bonus with sparse rewards, and more generally, demonstrating the advantage of prioritising exploration in planning, by using Boltzmann search policies.

An interesting area of future work may include investigating good heuristics for setting parameters in BTS and DENTS. We noticed that the best value for the search temperature tended to be the same order of magnitude as the optimal value at the root node $V^*(s_0)$, which suggests a heuristic similar to [21] may be reasonable.

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
