# OpenReview forum: "Monte Carlo Tree Search with Boltzmann Exploration"
_NeurIPS.cc/2023/Conference — NeurIPS 2023 poster_

### Official Review · Reviewer_f8iz · 2023-06-30

**Soundness:** 3 good
**Presentation:** 3 good
**Contribution:** 3 good
**Rating:** 6
**Confidence:** 4

**Summary:**

This paper introduces two variants of MCTS designed to perform more exploration during search than the commonly used UCT search policy, while resolving an issue with a previously introduced MCTS variant, Maximum ENtropy Tree Search (MENTS), which was designed to explore more than UCT, but which does not converge to the optimal policy / value function with an increasing number of MCTS trials.

To address this issue, the authors modify MCTS in two different ways. In the first variant, Boltzmann Tree Search (BTS), the authors use a Boltzmann search policy during the selection/search phase of MCTS, instead of using UCT, but perform a regular Bellman backup to compute state and action values. In the second variant, Decaying ENtropy Tree Search (DENTS), they modify MENTs to separately keep track of the policy entropy and the (unmodified) value functions, allowing them to slowly decay the entropy contribution over time, while using the entropy-augmented value functions to drive exploration during search. Both variants of MCTS are consistent -- i.e. they value function they compute for the root node converges to the optimal value as the number of MCTS trials increases, and the simple regret decays exponentially with respect to the number of MCTS trials.

The authors evaluate BTS and DENTS against baseline methods on both toy MDP problems, and in a 9x9 Go round-robin tournament. They find that BTS and DENTS perform similarly or slightly better than existing baselines on the toy problems, and consistently outcompete other methods (modified to make use of neural-networks) in the Go tournament.

**Strengths:**

- The paper clearly explains an existing problem with MENTS not converging to the optimal root-node policy with increasing trials  (e.g. using the D-chain problem to illustrate the point).
- The "fix" that this paper proposes to MENTS is intuitive and clearly explained.
- The theoretical results appear to be sound, with extensive proofs (though I did not check these carefully).
- The empirical results demonstrate that BTS & DENTS can improve upon baseline search policies like UCT and MENTS in non-trivial problems (i.e. Go), while generally performing no worse than the baselines on simpler problems.

**Weaknesses:**

- The motivation for increasing exploration during planning/search could be better explained.
  - In particular, I think the authors could better highlight that UCT was justified in the context of selecting actions in the real-world, and that per Tolpin & Shimony et al [1, 2], using UCT in *simulation* doesn't make as much sense, since computational actions are less costly than actual actions.
  - This better motivates the use of a Boltzmann search policy, since exploring more when "thinking" could ultimately lead to better exploitation when finally acting / executing.
  - Similarly, the authors could better explain their choice of using simple regret (instead of cumulative regret) as the metric for an MCTS algorithm, following the discussion / reasoning in [1, 2]. Cumulative regret doesn't matter if the cost of computation is effectively zero (and only matters slightly if computation cost is much smaller the actual cost of acting).

- I don't think this should count very much against the paper, given the extensive effort at formally proving the stated results (which I think is valuable, and should be recognized), but it's arguably the case that BTS are DENTS are fairly incremental and straightforward extensions of MENTS.

[1] David Tolpin and Solomon Shimony. Mcts based on simple regret. In Proceedings of the AAAI Conference on Artificial Intelligence, volume 26, pages 570–576, 2012.

[2] Nicholas Hay, Stuart Russell, David Tolpin, and Solomon Eyal Shimony. Selecting computations: Theory and applications. arXiv preprint arXiv:1408.2048, 2014.

**Questions:**

- Why does the search temperature $\alpha$ in DENTS differ from the entropy weighting term $\beta$? By default (e.g. in MENTS) this should be the same -- is there a reason why it makes sense have them vary separately? Why not decay $\alpha$ with increasing node visits as well?

- In the AR version of DENTS used for the Go experiments, both $\alpha$ and $\beta$ are decayed. AR-MENTS is then described as a variant of AR-DENTS, where $\alpha$ is set equal to $\beta$. But in AR-MENTS, are $\alpha$ and $\beta$ set to a fixed value, or decayed with increasing node visits? I'm assuming it's the former, in which case that should be clarified in the Appendix, but if it's the latter, how is AR-MENTS different from AR-DENTS?

- What is the value of DENTS over BTS, or vice versa? Are their high-level rules for when one should be preferred over the other?

Minor comments:
- There seems to be some inconsistency in notation in equations 11 and 19 vs. the text surrounding them -- $N(s)$ and $N(s, a)$ are used in the equations, but the text uses $m_s$ and $m_a$.

Post-Rebuttal Response:

The author's have addressed my questions, and provided additional experiments showing how their proposed methods compare favorably against existing baselines. I continue to recommend acceptance.

**Limitations:**

One benefit of computing the max entropy policy, as MENTS does, is that it might help with exploration when *acting* (not just planning) --- this is perhaps helpful when the agent doesn't have an accurate model of the world, and needs to learn it over time. Adding some discussion on this point would be good, since it seems like a potential limitation of

Apart from that, the authors have addressed the limitation of BTS and DENTS that the require $O(|\mathcal{A}|)$ backups relative to UCT, which may hinder performance when actions spaces are large.

---

> ### Author Rebuttal · Authors · 2023-08-09
>
> Thank you for taking the time to review and providing us with helpful comments.
>
> > The motivation for increasing exploration during planning/search could be better explained.
>
> Thank you for the suggestions on clarifying the motivations for using simple regret and increasing exploration. We will make sure to update the paper with language similar to the explanation we gave to reviewer nux6 with the help of your review.
>
> > Why does the search temperature $\alpha$ in DENTS differ from the entropy weighting term $\beta$? By default (e.g. in MENTS) this should be the same -- is there a reason why it makes sense have them vary separately? Why not decay $\alpha$ with increasing node visits as well?
>
> In DENTS we separated the $\alpha$ and $\beta$ parameters so that the temperature of the Boltzmann search policy can be separated from the weighting of the entropy objective. Firstly, we want to decay the value of $\beta$ to avoid the parameter sensitivity problems that MENTS suffers from in the modified D-chain and Frozen Lake environments. Specifically, if $\beta$ is not decayed, then we risk prioritising entropy more in the search policy than the standard objective (as can be seen in Figure 26 in the appendix). In contrast, having $\alpha > 0$ is a necessary condition for the exponential simple regret bound so that all actions are always explored with some minimum probability. We could let $\alpha\rightarrow 0$ in DENTS and it would be *consistent* (as per the definition in lines 153/154 of the paper), but it would not achieve a similar simple-regret bound to BTS. We could have alternatively allowed $\alpha\rightarrow\alpha_{\min}$, but we decided to keep $\alpha$ fixed for a cleaner presentation.
>
> > In the AR version of DENTS used for the Go experiments, both $\alpha$ and $\beta$ are decayed. AR-MENTS is then described as a variant of AR-DENTS, where $\alpha$ is set equal to $\beta$. But in AR-MENTS, are $\alpha$ and $\beta$ set to a fixed value, or decayed with increasing node visits? I'm assuming it's the former, in which case that should be clarified in the Appendix, but if it's the latter, how is AR-MENTS different from AR-DENTS?
>
> You are correct in assuming the former. For AR-MENTS, $\beta(m)=\alpha(m)=\alpha_{fix}$ for some fixed value $\alpha_{fix}$. We will clarify.
>
> > What is the value of DENTS over BTS, or vice versa? Are their high-level rules for when one should be preferred over the other?
>
> BTS will outperform DENTS when the reward signal is more dense and tailored to drive exploration since doesn’t have the additional computational overhead of the entropy backup. Note that in complex environments this often requires reward engineering where specifying an appropriate reward is a difficult task in of itself versus using entropy for exploration.
>
> (Repeated from response to reviewer yYGy)
> DENTS will outperform BTS when entropy is a useful exploration bonus, such as in cases where rewards are sparse and Frozen Lake is a good example of this. In Frozen Lake states that correspond to falling down a hole have an entropy of 0, and so the entropy bonus encourages DENTS (and other entropy algorithms) to explore paths that don’t fall into the holes. This benefit can be seen in Figure 3a, where BTS is slower to converge than DENTS and MENTS.
>
> > One benefit of computing the max entropy policy, as MENTS does, is that it might help with exploration when acting (not just planning) --- this is perhaps helpful when the agent doesn't have an accurate model of the world, and needs to learn it over time. Adding some discussion on this point would be good, since it seems like a potential limitation of
>
> Thank you for pointing this out, we will make sure to add this as an additional limitation. Our algorithms may not be useful when converging to the maximum entropy policy is desirable. As you mention, exploration while acting will be useful when the agent does not have an accurate model of the world and still needs to learn it, and MENTS would be more appropriate in these scenarios.

---

> > ### Comment · Reviewer_f8iz · 2023-08-12
> > **Thanks for the response.**
> >
> > Thank you for responding to my questions, which helped to clarify my understanding of the paper. I have read the rebuttal and will be maintaining my recommendation for acceptance.

---

### Official Review · Reviewer_yYGy · 2023-07-03

**Soundness:** 3 good
**Presentation:** 3 good
**Contribution:** 3 good
**Rating:** 5
**Confidence:** 3

**Summary:**

This paper proposes two new algorithms, Boltzmann Tree Search (BTS) and Decaying Entropy Tree Search (DENTS), based on the Maximum Entropy Tree Search (MENTS) method. The authors prove that in certain situations, MENTS fails to converge to the optimal policy. They analyze BTS and DENTS using simple regret and show that these algorithms will recommend the optimal actions. Finally, the authors evaluate various MCTS algorithms, where BTS and DENTS significantly outperform other MCTS algorithms.

**Strengths:**

1. The paper is well-structured, with clear explanations of the concepts and methods used.
2. The paper identifies the limitations of MENTS and proposes two new algorithms, BTS and DENTS, to address the issues of MENTS. The authors conduct theoretical analysis and perform experiments to validate their proposed solutions.
3. The experiments show that the proposed algorithms outperform other MCTS algorithms.

**Weaknesses:**

It is not quite convincing for the baseline program used in the Go experiments. The Komi in 9x9 Go is usually set to 7 (a balanced setting for both players). Black will have an advantage when Komi is set to 6.5. In addition, in Table 1, does PUCT have the same performance (strength) as KataGo? Why not compare the AR-* methods to KataGo directly? Besides, it is quite confusing why the author uses 128 CPU threads. To the best of my knowledge, this is not the best machine configuration for running KataGo. Using 128 CPU threads may even hurt the performance of KataGo. Moreover, in line 266, the authors said they added Dirichlet noise at the root node. In AlphaGo Zero, there is no need to add Dirichlet noise during evaluation.

**Questions:**

1. The authors mention that MENTS may struggle to converge under certain specific circumstances by using the D-Chain as an example. However, it is unclear whether this phenomenon generally happens in other environments.

2. The author mentions the difficulty of selecting β. The experiments also show that BTS (DENTS with β=0) performs better than DENTS. Do the authors conduct any analysis or have any suggestions for β? In what scenarios where DENTS will perform better than BTS?

3. The author claims that DENTS will be similar to MENTS when setting β(m) = α, is there any experimental/theoretical explanation for this part? The relationship is a bit unclear from the mathematical formula.

4. How does MCTS operate in a stochastic environment for Sailing problems?

5. Why were RENTS and TENTS only tested on the 6x6 Sailing Problem and not on the 8x12 Frozen Lake?

Other minor suggestions:
1. The lines in Figure 2 are overlapping, making it difficult to see clearly.
2. In equation (4), should it be $V_{\text{sft}}^{\pi}(s', t)$ or $V^{\pi}_{\text{sft}}(s', t + 1)$?

**Limitations:**

The authors addressed their limitations in section 5.3.

---

> ### Author Rebuttal · Authors · 2023-08-09
>
> Firstly, thank you for your review and comments to help us improve our paper.
>
> > The Komi in 9x9 Go is usually set to 7 (a balanced setting for both players). Black will have an advantage when Komi is set to 6.5.
>
> We chose a komi of 6.5 because it is commonly used by humans in 9x9 Go and a half-integer komi ensures all games were decisive. Black will have an advantage in 9x9 Go with a komi of 6.5 and the comparison is still fair as each algorithm is given an equal number of games as black and white.
>
> > In addition, in Table 1, does PUCT have the same performance (strength) as KataGo? Why not compare the AR-* methods to KataGo directly?
>
> We didn’t compare the AR-* algorithms directly to KataGo because our paper is about general purpose search algorithms. To ensure the evaluation is implementation independent we used a common implementation between our MCTS algorithms. KataGo uses a variety of heuristics on top of standard PUCT [1] and is a domain-specific library that has been optimised for Go. Our work does not intend to be a new state-of-the-art for Go, and KataGo will beat our algorithms given the same time constraints. Our algorithms could be incorporated into and potentially improve an MCTS agent such as KataGo. However, this is beyond the scope of this paper, which proposes a general-purpose planning approach.
>
> > Besides, it is quite confusing why the author uses 128 CPU threads. … In AlphaGo Zero, there is no need to add Dirichlet noise during evaluation.
>
> We ran `./katago benchmark` and it recommended 32 search threads for KataGo. In Table 1 in our author rebuttal, we also found that the best number of threads to use for PUCT was 32.
>
> When updating our results (Table 3 of the author rebuttal), we ran PUCT with and without adding Dirichlet noise. In each match, the agent with Dirichlet noise won more games so we have kept those results, suggesting that considering novel moves was still useful. We also ran PUCT+Dirichlet noise against PUCT, and the agent adding noise won with a score of 28-23.
>
> > The authors mention that MENTS may struggle to converge under certain specific circumstances by using the D-Chain as an example. However, it is unclear whether this phenomenon generally happens in other environments.
>
> MENTS parameter sensitivity can also be demonstrated in the Frozen Lake environment, as can be seen in Figure 26 in the appendix. This figure includes BTS and DENTS running with the same temperatures for reference.
>
> > The author mentions the difficulty of selecting β. The experiments also show that BTS (DENTS with β=0) performs better than DENTS. Do the authors conduct any analysis or have any suggestions for β? In what scenarios where DENTS will perform better than BTS?
>
> In our experiments, we typically set $\beta(m)=\alpha/\log(e+m)$ to reduce the search space over hyperparameters. The best way to set $\beta$ will depend on the environment. Currently, as a rule of thumb, we recommend using $\beta(m)=\beta_{\text{init}}/\log(e+m)$ and then tuning the value of $\beta_{\text{init}}$ for the specific environment. If a large amount of exploration is desired then $\beta_{\text{init}}$ can be set to a large value, while still allowing $\beta$ to decay to zero at the root of the tree and encourage exploration in deeper parts of the tree. Sometimes entropy may be disadvantageous, such as in our Go experiments, in which case using BTS ($\beta(m)=0$) performs better.
>
> DENTS will outperform BTS when entropy is a useful exploration bonus, such as in cases where rewards are sparse and Frozen Lake is a good example of this. In Frozen Lake states that correspond to falling down a hole have an entropy of 0, and so the entropy bonus encourages DENTS (and other entropy algorithms) to explore paths that don’t fall into the holes. This benefit can be seen in Figure 3a, where BTS is slower to converge than DENTS and MENTS.
>
> > The author claims that DENTS will be similar to MENTS when setting β(m) = α, is there any experimental/theoretical explanation for this part? The relationship is a bit unclear from the mathematical formula.
>
> Figure 1 in the author rebuttal empirically demonstrates the similarity of running DENTS with $\beta(m)=\alpha$ and MENTS on the D-chain and Frozen Lake environments. A theoretical relationship may be possible as Lemma 4 and Corollary 5 from [2] suggests that optimal soft values can be written as a sum of an optimal soft Q-values and an entropy term, similar to the “soft values” we use in DENTS.
>
> > How does MCTS operate in a stochastic environment for Sailing problems?
>
> MCTS handles stochastic environments by using decision and chance nodes. Decision nodes correspond to states, chance nodes correspond to state-action pairs, and child nodes are selected by sampling from the transition distribution $p(s’|s,a)$. The work in [3] is a good reference for the use of MCTS for planning under uncertianty.
>
> > In equation (4), should it be t or t+1?
>
> It should be $t+1$, thank you.
>
> > Why were RENTS and TENTS only tested on the 6x6 Sailing Problem and not on the 8x12 Frozen Lake?
>
> (Repeated from response to reviewer nux6)
> We did run RENTS and TENTS on Frozen Lake, the results can be seen in Figure 29 in the appendix. It was an oversight to not include them in the main paper, and we will update Figure 3 to include these results and present them more clearly. RENTS and TENTS performed similarly to MENTS and DENTS, which is in line with the maximum entropy objective (and its variants) being a useful exploration bonus in this sparse reward environment.
>
> [1] https://github.com/lightvector/KataGo/blob/master/docs/KataGoMethods.md
>
> [2] Nachum, Ofir, et al. "Bridging the gap between value and policy based reinforcement learning." Advances in neural information processing systems 30 (2017)
>
> [3] Keller, Thomas, and Malte Helmert. "Trial-based heuristic tree search for finite horizon MDPs." Proceedings of the International Conference on Automated Planning and Scheduling. Vol. 23. 2013

---

> > ### Comment · Reviewer_yYGy · 2023-08-16
> >
> > Thanks for your response. I have a few questions and suggestions.
> >
> > It is weird that using Dirichlet noise performs better than without Dirichlet noise during evaluation. As AlphaZero didn’t use Dirichlet noise in the evaluation.
> >
> > > Our work does not intend to be a new state-of-the-art for Go, and KataGo will beat our algorithms given the same time constraints.
> >
> > I understand that you are not trying to beat KataGo. However, it is still important that compare your baseline model (PUCT) to KataGo to verify your methods. In any case, as you used the same neural network as KataGo, your PUCT should at least has a comparable strength to KataGo even without these heuristics.

---

> > > ### Author Response · Authors · 2023-08-18
> > >
> > > Thank you for your suggestions
> > >
> > > We agree that we should compare our PUCT agent with KataGo to verify its strength, we were in the process of setting up these experiments this week. We have now run our PUCT agent versus Kata Go and limited them to 1600 trials to give an idea of relative strength. On the 9x9 board, the score was 61(PUCT) - 39(KataGo) and on the 19x19 board, the score was 35(PUCT) - 65(KataGo). We were a little surprised to see our agent win in 9x9, but it could be that KataGo's alterations to UCT are more suited to the 19x19 board, or because our implementation runs slower (so when selecting moves the values used in UCB are likely more up to date)
> > >
> > > We also agree that adding Dirichlet noise seems unnecessary after you pointed it out in your review. While updating the results when we update our paper, we will continue to try with and without the noise to make sure our PUCT agent is as strong as possible. Given the results from our PUCT agent versus KataGo, it could be that the noise is sometimes useful in 9x9 Go with fewer trials but not so much in 19x19 Go

---

> > > > ### Comment · Reviewer_yYGy · 2023-08-21
> > > >
> > > > Good to see these additional experiments. I would suggest including these experiments for clarity of the baseline (PUCT) model. I have adjusted my score accordingly.

---

> > > > > ### Author Response · Authors · 2023-08-21
> > > > >
> > > > > Thank you, we completely agree and plan to add these experiments from our PUCT agent versus KataGo into the paper to give readers and idea of the strength of our PUCT agent

---

### Official Review · Reviewer_gH3c · 2023-07-05

**Soundness:** 2 fair
**Presentation:** 3 good
**Contribution:** 2 fair
**Rating:** 5
**Confidence:** 3

**Summary:**

<< I have read the authors' rebuttal and have raised my score based on the discussion >>

The paper addresses challenges in online tree-based search algorithms, such as UCT, which have shortcomings such as being prone to local optima and a slower convergence rate to the optimal action. An alternative approach, Maximum ENtropy Tree Search (MENTS), despite theoretically optimal performance, has practical limitations like sensitivity to the temperature parameter, a tendency to over explore, and increased computational expense. The authors present two new algorithms to address these issues: Boltzmann Tree Search (BTS) and Decaying ENtropy Tree Search (DENTS). BTS employs a Boltzmann search policy, focusing solely on reward maximisation, whereas DENTS introduces entropy backups that can be used with varying weighting in the search. Both algorithms preserve the exploratory benefits of MENTS, and importantly, they consistently converge to the optimal reward maximising policy. Key contributions of the paper include highlighting the misalignment of the entropy objective in MENTS with reward maximisation, introducing BTS and DENTS, providing an analysis of MENTS, BTS, and DENTS through the lens of simple regret, and demonstrating the improved performance of BTS and DENTS over other MCTS algorithms in specific environments and games.

**Strengths:**

Novelty: The presented Boltzmann Tree Search (BTS) algorithm is a variant of the Maximum ENtropy Tree Search (MENTS), substituting the soft update value in equation 13 with the max operator in equation 19. The Decaying ENtropy Tree Search (DENTS) proposes a novel method for decaying exploration to ultimately converge to the optimal solution, an aspect MENTS struggles with due to its focus on maintaining entropy in the search policy. The novelty of this paper's contributions, while discernible, seems to be limited.

Clarity: While the paper is generally well-written, comprehension is somewhat impeded by the multitude of mathematical symbols and technical jargon used throughout. An effort to streamline and elucidate the language could significantly enhance readability.

Significance: The paper's experimental results highlight DENTS' robustness to the adjustment of the temperature hyperparameter, a critical element in MENTS. Moreover, the simplicity of the proposed changes augments their potential for practical adoption in the field.

**Weaknesses:**

W1: The novelty of the work seems somewhat limited. It remains unclear whether the improvements result from the incorporation of entropy in the case of DENTS or from the use of hard updates instead of soft updates in BTS.

W2: The paper emphasizes the sensitivity of MENTS to the temperature hyperparameter. It is true that many online search algorithms rely on hyperparameter tuning for optimal performance. However, it is noteworthy that MENTS outperforms both BTS and DENTS on the 10-chain and gridworld tasks when the temperature hyperparameter is appropriately tuned. Therefore, it could be argued that the only advantage of using BTS or DENTS is their lesser dependency on temperature. Yet, with the correct tuning of this singular parameter, MENTS could potentially offer a superior solution.

**Questions:**

Q1: I found it somewhat surprising that MENTS performed relatively poorly on the game of Go, especially considering its strong performance in other domains. Can you confirm whether the temperature hyperparameter for MENTS was appropriately tuned in this case?

Q2: What methodologies were employed by the authors to tune the hyperparameters for different baseline algorithms, including BTS and DENTS?

Q3: To better understand the influence of hyperparameters on BTS and DENTS and gauge their robustness, would it be possible for the authors to compare their performance for varying values of the alpha parameter?

**Limitations:**

The authors discussed the limitations of their work. No discussion needed regarding potential negative societal impact.

---

> ### Author Rebuttal · Authors · 2023-08-09
>
> Firstly, thank you for reviewing our work and giving us valuable feedback.
>
> > Q3: To better understand the influence of hyperparameters on BTS and DENTS and gauge their robustness, would it be possible for the authors to compare their performance for varying values of the alpha parameter?
>
> We varied the temperature parameters of all of the algorithms on an 8x8 Frozen Lake environment (Figure 5a) in Figures 26, 27 and 28 in Appendix C.2.2. These figures show that for high temperatures in Frozen Lake, MENTS and TENTS fail to find the optimal paths, while BTS and DENTS for the same temperatures do find the optimal paths. RENTS did not suffer from the same parameter sensitivity in Frozen Lake, however, it also behaved similarly to UCT (i.e. it never found the reward of 1.0) on the D-chain environments as can be seen in Figures 10 and 16 from Appendix C.2.1 (the corresponding results for UCT are given in Figures 8 and 14).
>
> > W1: The novelty of the work seems somewhat limited. It remains unclear whether the improvements result from the incorporation of entropy in the case of DENTS or from the use of hard updates instead of soft updates in BTS.
>
> We believe there are advantages to using Boltzmann search policies in MCTS, as they allow for greater exploration during planning, and are able to achieve better simple regret bounds than UCT [1]. Thus, theoretically they can provide better recommendations.
>
> Using entropy encourages exploration and for example can be advantageous in domains with sparse rewards such as Frozen Lake. However, it can also hinder performance in some environments, as can be seen in our results for Go where BTS outperformed the other algorithms. However, BTS will outperform DENTS when the reward signal is more dense and tailored to drive exploration without needing entropy, since BTS doesn’t have the additional computational overhead of the entropy backup. As such, it is useful to utilise the advantages of Boltzmann search policies, while having the option whether to use an entropy objective or not, depending on how informative the reward signal is.
>
> > W2: … However, it is noteworthy that MENTS outperforms both BTS and DENTS on the 10-chain and gridworld tasks when the temperature hyperparameter is appropriately tuned. Therefore, it could be argued that the only advantage of using BTS or DENTS is their lesser dependency on temperature. Yet, with the correct tuning of this singular parameter, MENTS could potentially offer a superior solution.
>
> In cases where MENTS would outperform BTS, we can tune DENTS to match the performance of MENTS. Specifically, we can set $\beta(m)=\alpha$ in DENTS, so that the search policy is similar to the MENTS search policy, while still using Bellman values for recommendations. See Figure 1 in the rebuttal pdf to see DENTS running under this configuration versus MENTS on the D-chain and Frozen Lake environments.
>
> > Q2: What methodologies were employed by the authors to tune the hyperparameters for different baseline algorithms, including BTS and DENTS?
>
> To tune the algorithms on the gridworld environments, we performed a grid-search over the UCT bias, search temperature $\alpha$ and MENTS exploration parameter $\epsilon$ that were relevant for each algorithm. For DENTS we set $\beta(m)=\alpha/\log(e+m)$. In Frozen Lake the parameters that achieved the best value after 500k trials on the 8x12 Frozen Lake from Figure 5b were selected for the test environment from Figure 5c. In the Sailing problem we performed the same routine, using an initial wind direction of North for parameter selection and initial wind direction of South-East for evaluation. Full details are given in Appendix C.4.
>
> In the Go experiments, we tuned parameters using round-robin tournaments varying one parameter at a time. We selected the value that won the most matches (a match being the combined score playing an equal number of games as black and white) and broke ties using the number of games won over the tournament. Full details are given in Appendix C.5.
>
> > Q1: I found it somewhat surprising that MENTS performed relatively poorly on the game of Go, especially considering its strong performance in other domains. Can you confirm whether the temperature hyperparameter for MENTS was appropriately tuned in this case?
>
> We followed the methodology outlined above for tuning MENTS temperature for Go. However, the temperature selected in the paper was the smallest value that we tried, suggesting that lower values might perform better. We have since run more games to tune the MENTS hyperparameter (Table 2 in author rebuttal) and have updated the results for 9x9 Go (Table 3 in author rebuttal). After updating the MENTS temperature parameter it still lost all of its matches, with its closest result being a 31-69 loss to TENTS. MENTS migt not perform as well on Go because often in games there will be *critical moves* that an agent needs to make and entropy may encourage prioritising moves that are not the critical move. Full details on hyperparameter selection for the Go experiments can be found in Appendix C.5.2.
>
> [1]  Sébastien Bubeck, Rémi Munos, and Gilles Stoltz. Pure exploration in multi-armed bandits problems. In International conference on Algorithmic learning theory, pages 23–37. Springer, 2009.

---

> > ### Comment · Reviewer_gH3c · 2023-08-19
> > **Further clarifications**
> >
> > Thank you for addressing the points I raised in my review. Your clarifications are appreciated. Nevertheless, I still have some lingering questions about the scenarios in which BTS and MENTS significantly surpass existing methods.
> >
> > Q1. Can you elaborate on the specific properties of problems that make them more amenable to BTS or MENTS?
> >
> > Q2. It would be beneficial to have a concise table comparing the strengths and weaknesses of MENTS, DENTS, and BTS. For instance, you highlighted reward signal density in your rebuttal. This would help in providing a more clear-cut comparison.
> >
> > Q3. Lastly, how would BTS and DENTS stack up against a method that employs Thompson sampling in MCTS, as presented in Bai et.al. [1]? It's noteworthy that Thompson Sampling also boasts superior regret convergence compared to UCT.
> >
> >
> > [1] Bai et.al., Bayesian Mixture Modelling and Inference based Thompson Sampling in Monte-Carlo Tree Search.

---

> > > ### Author Response · Authors · 2023-08-20
> > >
> > > > Can you elaborate on the specific properties of problems that make them more amenable to BTS or MENTS?
> > >
> > > Two properties of environments that would make them more amenable to using entropy for exploration are large delayed rewards, and, containing trap states (i.e. states where the agent cannot move to another state for the rest of the trial). The Frozen Lake environment demonstrates both of these properties, there is a large reward that the agent only gets when it reaches the goal, and there are holes that the agent falls into.
> > >
> > > It may also be the case that there are additional properties that we haven’t considered or thought of yet.
> > >
> > > We also note that the environment that we considered with these properties (Frozen Lake) was also the environment that exhibited the parameter sensitivity in Figure 26 in the appendix.
> > >
> > > When the reward signal is dense or engineered so that optimal policy can be found without needing additional exploration, then using entropy for additional exploration may cause unnecessary additional exploration, in which case BTS would be more suitable.
> > >
> > > > It would be beneficial to have a concise table comparing the strengths and weaknesses of MENTS, DENTS, and BTS. For instance, you highlighted reward signal density in your rebuttal. This would help in providing a more clear-cut comparison.
> > >
> > > We agree that it would help with clarity, here is a table comparing the different algorithms, and we will include this table in the appendix when we revise the paper.
> > > |  | UCT | MENTS | BTS | DENTS |
> > > |---|---|---|---|---|
> > > | Is consistent (will converge to a simple regret of 0 as $n \rightarrow \infty$) for any setting of hyperparameters | Yes | No | Yes | Yes |
> > > | Can use entropy for exploration (e.g. helpful for sparser rewards) | No | Yes | No | Yes |
> > > | Uses Boltzmann policies for exploration (stochastic action selection) | No | Yes | Yes | Yes |
> > > | Optimises for cumulative regret / penalises suboptimal actions during planning | Yes | No | No | No |
> > > | Optimised (amortised) complexity to run $n$ trials | $O(nHA)$ | $O(n(H+A))$ |  $O(n(H\log(A)+A))$ | $O(n(H\log(A)+A))$ |
> > >
> > > We also note that using Boltzmann policies has multiple other advantages, such as being able to use the alias method for sampling to make action selection take amortised $O(1)$ time, and the stochastic sampling of actions often leads to higher utilisation of threads when running a multi-threaded tree search.
> > >
> > > With respect to the complexities, since we discovered that we can use the Alias method for sampling, we considered how efficiently backups can be implemented as that was now the bottleneck. The backups from Equations (12), (13), (18), (23) and (24) from the paper can be implemented in $O(1)$ because only one child value changes. Computing the $\max$ backup from equation (19) takes $O(\log(A))$. Additionally, we could improve on the complexity of BTS and DENTS by approximating the $\max(\cdot)$ operation with a $\tau \log \sum \exp (\cdot / \tau)$ with a suitably small $\tau$, which would match the optimised MENTS complexity of $O(n(H+A))$.
> > >
> > > > Lastly, how would BTS and DENTS stack up against a method that employs Thompson sampling in MCTS, as presented in Bai et.al. [1]? It's noteworthy that Thompson Sampling also boasts superior regret convergence compared to UCT.
> > >
> > > We are confident that using DNG-MCTS from Bai et.al. would still struggle with the same issues as UCT. Because it is also designed with cumulative regret in mind, similarly to UCT, it moves towards greedy exploiting policies as more trials are run. For example, if we ran DNG-MCTS on the Frozen Lake environment (note that rewards and actions are deterministic), then the posterior distributions will tend towards degenerate distributions and action selection will tend towards greedy action selection. That is, it would not explore sufficiently to find better paths from the starting location to the goal. BTS, DENTS and MENTS outperform UCT on the Frozen Lake environment because they emphasise exploration more, which is necessary to find improved paths to the goal.

---

> > > > ### Comment · Reviewer_gH3c · 2023-08-21
> > > >
> > > > Thank you for addressing my concerns. I believe it would be beneficial to mention the specific properties of environments in the experiment or discussion section that are more suitable for BTS/DENTS. I have adjusted my score in favor of acceptance.

---

> > > > > ### Author Response · Authors · 2023-08-21
> > > > >
> > > > > And thank you very much for the feedback and we will discuss the properties of environments above in the paper when we revise it

---

### Official Review · Reviewer_nux6 · 2023-07-07

**Soundness:** 3 good
**Presentation:** 3 good
**Contribution:** 4 excellent
**Rating:** 6
**Confidence:** 3

**Summary:**

The paper presents two MCTS based algorithms called  Boltzmann Tree Search (BTS) and Decaying ENtropy Tree-Search (DENTS). These algorithms overcome the limitation of two state-of-the-art algorithms pUCT and MENTS of under and over exploration respectively. Further, the paper provides simple “simple regret” analysis and shows state-of-the-art performance on benchmark tasks.

**Strengths:**

1. The paper is very well structured and easy to follow for the most part.
2. The empirical results are impressive, especially on Go.
3. The algorithms are novel to my knowledge and are inspiring.


**Weaknesses:**

1. The motivation for analyzing simple regret instead of cumulative regret is not clear. This limits the contribution of the proposed approach to some extent. (I will update the score post rebuttal, given the authors' response).
2. The paper puts all the theoretical proof in the appendix. Please provide at least a proof sketch in the main paper.
3. Section 6.1 is not clearly written. What do you mean by “ To utilize ... a terminal/goal state.” (lines 232-234)

Minor
1. Graphs in Figure 2 are of poor quality. Please consider using transparent lines or a variety of dashed lines (different patterns of dashes) to represent different algorithms.
2. Please increase the font size of the text in Figure 3.

Note: I haven't carefully taken a look at the proofs in the appendix. I might update my score based on correctness.


**Questions:**

1. Line number 18 - What do you mean by “exact methods”?
2. Why H-MCTS [27] (or other algorithms that focus on simple regret) is not used as a baseline?
3. Are there no approaches that study how to decrease alpha in MENTS so that Theorem 2-like statement holds?
4. How to set the value of alpha in BTS?
5. Why are RENTS and TENTS not compared on Frozen Lake?


**Limitations:**

Limitations are clearly mentioned in the paper.

---

> ### Author Rebuttal · Authors · 2023-08-09
>
> Firstly thank you for taking the time to review our work and provide helpful feedback.
>
> > The motivation for analyzing simple regret instead of cumulative regret is not clear. This limits the contribution of the proposed approach to some extent.
>
> Cumulative regret is more appropriate in situations where taking actions during the planning/learning stage corresponds to interacting with the real world, and hence there is a need to explicitly consider the cost accumulated by taking actions. However, our setting considers planning using a model or simulator, without any interaction with the real world during the planning stage. Simple regret is more appropriate in this setting, because it ignores the cost of simulated actions during planning, and only considers the cost of the action executed in the real world, once the planning stage is complete. By considering simple regret, algorithms are not penalised for under-exploiting during the planning stage and are thus able to explore more effectively, leading to better recommended actions.
>
> Please also note Reviewer f8iz’s comments on the importance of using simple regret in our work. They provided some suggestions for motivating its use similar to Tolpin & Shimony et al [1, 2], which we paraphrased above and will add to the paper.
>
> > The paper puts all the theoretical proof in the appendix. Please provide at least a proof sketch in the main paper.
>
> We will add short proof outlines to the paper, and make space for this by making the introduction and background sections more concise.
>
> > Section 6.1 is not clearly written. What do you mean by “ To utilize ... a terminal/goal state.” (lines 232-234)
>
> We wanted to make clear that we use horizons that are long enough for the trials to terminate because the agent either reached the goal or fell into a hole, rather than terminating because the max horizon length was reached. We used planning horizons of 100 and 50 on the Frozen Lake and Sailing environments, respectively. We will edit the paper to make this clearer.
>
> > Line number 18 - What do you mean by “exact methods”?
>
> We were referring to methods that compute a full policy, such as value iteration. We will update the wording to instead specify methods that compute a full policy.
>
> > Why H-MCTS [27] (or other algorithms that focus on simple regret) is not used as a baseline?
>
> We will update the paper to include H-MCTS in the evaluation. To the best of our knowledge, prior MCTS algorithms designed for simple regret, such as H-MCTS [3] and BRUE [4], are adapted from UCT. As such we believe that our empirical results using UCT still give a good indication of how these other algorithms would perform in practise.
>
> > Are there no approaches that study how to decrease alpha in MENTS so that Theorem 2-like statement holds?
>
> We actually tried decaying the temperature parameter in MENTS with respect to node visits. It is true that such an algorithm can achieve a similar simple regret bound to BTS and DENTS, but we found it didn’t work well in practise. To try and explain why: consider the maximum entropy objective from Equation (3) in the paper. Reducing the value of $\alpha$ changes the weighting of the entropy at that node in Equation (3). When the root node’s $\alpha$ becomes small, the search policy (Equation (11)) becomes increasingly greedy and stops exploring close to the root node. The root node’s child values $\hat{Q}^{N(s_0,a)}_{\text{sft}}(s_0,a)$ include values that were computed from backups with high values of $\alpha$ because they have been visited less times than the root. This becomes problematic because the algorithm stops exploring close to the root node, and selected greedily based on Q-values that were propagated from nodes still considering high values of $\alpha$. It was the issues with this algorithm that motivated our design of DENTS, to explicitly compute separate Bellman and (subtree) entropy values and decay the weighting on the entropy term.
>
> > How to set the value of alpha in BTS?
>
> $\alpha$ is a hyperparameter that needs to be tuned to get the best performance out of BTS, similar to the temperature in MENTS and the bias parameter in UCT. Across our experiments, the best value of alpha tended to be in the range of $V^*(s_0)/2$ to $V^*(s_0)/10$, so it may be possible set $\alpha=\hat{V}(s)/2$ for example to scale $\alpha$ using value estimates in the tree similar to [5].
>
> > Why are RENTS and TENTS not compared on Frozen Lake?
>
> We did run RENTS and TENTS on Frozen Lake, the results can be seen in Figure 29 in the appendix. It was an oversight to not include them in the main paper, and we will update Figure 3 to include these results and present them more clearly. RENTS and TENTS performed similarly to MENTS and DENTS, which is in line with the maximum entropy objective (and its variants) being a useful exploration bonus in this sparse reward environment.
>
> As suggested we will make the following updates to the paper:
> - Update the graphs in Figure 2 to use different line styles with markers to make the overlapping lines more clear
> - Increase the font sizes in figures for clarity
>
> [1] David Tolpin and Solomon Shimony. Mcts based on simple regret. In Proceedings of the AAAI Conference on Artificial Intelligence, volume 26, pages 570–576, 2012
>
> [2] Nicholas Hay, Stuart Russell, David Tolpin, and Solomon Eyal Shimony. Selecting computations: Theory and applications. arXiv preprint arXiv:1408.2048, 2014
>
> [3] Tom Pepels, Tristan Cazenave, Mark HM Winands, and Marc Lanctot. Minimizing simple and cumulative regret in monte-carlo tree search. In Workshop on Computer Games, pages 1–15. Springer, 2014
>
> [4] Zohar Feldman and Carmel Domshlak. Simple regret optimization in online planning for markov decision processes. Journal of Artificial Intelligence Research, 51:165–205, 2014
>
> [5] Thomas Keller and Patrick Eyerich. Prost: Probabilistic planning based on uct. In Twenty-Second International Conference on Automated Planning and Scheduling, 2012

---

> > ### Author Response · Authors · 2023-08-17
> > **HMCTS Results**
> >
> > Hi, we've now added HMCTS to our smaller experiments and can include it in our results. For selecting hyperparameters we varied the budget threshold (to switch from sequential halving to UCT) and the UCT bias parameters. For the UCT bias parameter we used searched over the same values as when we ran UCT independently.
> >
> > On the 10-chain HMCTS always recommends taking the reward of 0.9 for all of the values of budget threshold that we tried {1,3,10,30,100,300,1000}, and so it behaved identically to UCT here. This is because it is still using sample averages to make its recommendations, and the average return from the trials that search down the chain is lower than 0.9.
> >
> > On the Frozen Lake HMCTS also performed similarly to UCT and struggled to explore to find better paths to the goal. The only difference was that HMCTS managed to find its best path in fewer trials than UCT. This would likely happen because the use of sequential halving helps in the upper portion of the trees (for example eliminating actions that directly fall down holes), but ultimately it still relies on UCT and struggles with the same problems.
> >
> > In the sailing domain UCT already performs well and there wasn't any noticeable difference between HMCTS and UCT. We tried the values of {10,30,100,...,30000,100000} for the budget threshold in the gridworld domains.
> >
> > In conclusion, HMCTS's behaviour is very similar to UCT's in our experiments.

---

> > > ### Comment · Reviewer_nux6 · 2023-08-17
> > > **response to the rebuttal**
> > >
> > > Thanks for an elaborate response with the new results. Your response clears confusion on experimentation.
> > >
> > > 1) Can you provide a proof sketch during the rebuttal so that I and other reviewers can verify the correctness?
> > >
> > > 2) Now I am a bit more clear on simple regret vs cumulative regret, but I think you should perhaps explain this with an example, maybe in the appendix.

---

> > > > ### Author Response · Authors · 2023-08-19
> > > >
> > > > > Now I am a bit more clear on simple regret vs cumulative regret, but I think you should perhaps explain this with an example, maybe in the appendix
> > > >
> > > > We can also add an example to the appendix. The use of MCTS in robotics would provide a good example, such as using MCTS for path planning [1,2]. If the robot is equipped with a simulated environment to search in, then simple regret is would be a more appropriate metric, and if the robot can only interact in the real world then cumulative regret would be a more appropriate metric.
> > > >
> > > > [1] Janson, Lucas, Edward Schmerling, and Marco Pavone. "Monte Carlo motion planning for robot trajectory optimization under uncertainty." Robotics Research: Volume 2. Cham: Springer International Publishing, 2017. 343-361.
> > > >
> > > > [2] Dam, Tuan, et al. "Monte-Carlo robot path planning." IEEE Robotics and Automation Letters 7.4 (2022): 11213-11220.
> > > >
> > > > > Can you provide a proof sketch during the rebuttal so that I and other reviewers can verify the correctness?
> > > >
> > > > Here are some proof sketches for the results in the paper. We've tried to keep them short so that they can feasibly be put into the papers in a similar form. But please let us know if any more details would be helpful.
> > > >
> > > > ### Theorem 5.2
> > > >
> > > > Let $0<\pi_\min<\pi_\text{DENTS}(a|s)$ for all $s,a$, where $\pi_\min$ exists because $\exp(x)>0$ for any $x$ and $1/|\mathcal{A}|>0$ in Equation (21) (Lemma D.1). Then for a DENTS process $\Pr(E)\leq C_0\exp(-k_0N(s_t))$ iff $\Pr(E)\leq C_1\exp(-k_1N(s_t,a_t))$ iff $\Pr(E)\leq C_2\exp(-k_2N(s_{t+1}))$ holds for any event $E$ and constants $C_0,C_1,C_2,k_0,k_1,k_2$, which can be shown using $\pi_\min$ with Hoeffding bounds over indicator variables (Lemmas D.4, D.5).
> > > >
> > > > We can then show $\Pr\left(\left|\hat{Q}^{N(s_1,a)}(s_1,a) - Q^*(s_1,a)\right| > \varepsilon \right)\leq C_3\exp(-k_3\varepsilon^2n)$ by induction, where the base case of the induction $\hat{V}^{N(s_{H+1})}(s_{H+1}) = V^*(s_{H+1}) = 0$ holds vacuously (Lemmas D.10, D.15 and Theorem D.16). Let $\Delta_\mathcal{M}$ be a small constant (Equation (105)) such that $\forall a\in\mathcal{A}. \left|\hat{Q}^{N(s_1,a)}(s_1,a) - Q^*(s_1,a)\right| \leq \Delta_\mathcal{M}/2 \implies \psi_\text{DENTS}(s_1)=\pi^*(s_1)$. Then setting $\varepsilon=\Delta_\mathcal{M}/2$ gives a bound on $\Pr(\psi_\text{DENTS}(s_1)\neq\pi^*(s_1))$ which can be used in the definition of simple regret to give the result.
> > > >
> > > > ### Theorem 5.1
> > > >
> > > > Theorem 5.1 follows immediately from Theorem 5.2, as mathematically BTS is a special case of DENTS with $\beta(m)=0$.
> > > >
> > > > ### Theorem 4.2
> > > >
> > > > $\Pr\left(\left|\hat{Q_{\text{sft}}}^{N(s_1,a)}(s_1,a) - {Q_{\text{sft}}}^*(s_1,a)\right| > \varepsilon \right)\leq C_4\exp(-k_4\varepsilon^2n)$ can be shown similarly to Theorem 5.2. The restriction on the value of $\alpha$ is required, as similar to Theorem 5.2, we need to use $\left(\alpha\leq \Delta_\mathcal{M}/3H\log |\mathcal{A}|\right) \land \left(\forall a\in\mathcal{A} \left|\hat{Q_{\text{sft}}}^{N(s_1,a)}(s_1,a) - {Q_{\text{sft}}}^*(s_1,a)\right| \leq \Delta_{\mathcal{M}}/3\right) \implies \psi_\text{MENTS}(s_1)=\pi^*(s_1)$ to bound the probability of $\psi_{\text{MENTS}}(s_1)\neq \pi^*(s_1)$.
> > > >
> > > > ### Proposition 4.1
> > > >
> > > > This follows from computing the values of $Q^*_{\text{sft}}(1,a_1)$ and $Q^*_{\text{sft}}(1,a_2)$ in the modified 10-chain from Figure 1 of the paper using a temperature of $\alpha=1$. (Note that here $a_1,a_2$ refers to the two actions that can be taken in the D-Chain, rather than the variables representing the first and second actions taken on a trial. We will update the figure and paper to avoid this naming clash). We can use $\Pr\left(\left|\hat{Q_{\text{sft}}}^{N(s_1,a)}(s_1,a) - {Q_{\text{sft}}}^*(s_1,a)\right| > \varepsilon \right)\leq C_4\exp(-k_4\varepsilon^2n)$ from Theorem 4.2, to show $\Pr(\psi_\text{MENTS}(1) = a_1)\rightarrow 1$ as $n\rightarrow \infty$ below, so gives MENTS a simple regret that stays above 0 as $n$ tends to infinity.
> > > >
> > > > N.B. Original response on 19/8, edited on 20/8.

---

### Author Rebuttal · Authors · 2023-08-09

We have attached a pdf with updated results following comments from the reviewers.

In Figure 1 of the attached pdf we have updated results for DENTS running under the $\beta(m)=\alpha$ configuration in both the D-chain and Frozen Lake environments. These show empirically that DENTS can still match the performance of MENTS when MENTS is properly tuned.

We have run experiments to test the best number of threads to use in Go for PUCT and BTS (and other algorithms using Boltzmann policies), which can be found in Table 1. We tried using {16, 32, 64, 128} threads, and these results suggest that 32 threads were optimal for PUCT and that 128 was best out of the values tried for BTS.

Originally we tried temperature parameters from {1000, 500, 100, 50, 10} for all algorithms using Boltzmann policies (noting that rewards were scaled to be in the range [-100,100]). For MENTS and TENTS the best-performing temperature was 10, suggesting that the optimal temperature may be lower. Table 2 gives results from the updated tuning of the MENTS and TENTS temperatures, with the best-performing temperatures being 0.5 and 10 respectively. Note that the TENTS temperature remains unchanged at a value of 10, which beats all higher temperatures tried (Table 9 in paper appendix) and all lower temperatures (Table 2 in attached pdf).

In Table 3 we give updated results for MENTS and PUCT for the 9x9 Go round-robin. These matches were run giving all algorithms 32 threads and used the updated temperature of 0.5 for MENTS. The additional tuning of MENTS led it to win a few more games but ultimately the closest it came to winning a match was its 31-69 loss to TENTS. Similarly, the results for PUCT did not significantly change, BTS and DENTS won by a larger margin in these reruns but PUCT did draw with RENTS this time round.

In Table 4 of the attached pdf we give results on 19x19 Go (using the standard komi of 7.5 for 19x19 Go), which used the same parameters selected on 9x9 Go, and each agent was still given 15s to make a move. The only agent that was able to beat PUCT was BTS with a match score of 57-43. In 19x19 Go, PUCT was able to run a similar number of trials per move as BTS (\~5000), in contrast to the results in 9x9 Go from Table 1 of the paper where PUCT (\~13000) could not run as many trials as BTS (\~17000). Note that BTS is often able to run more trials because the emphasis on exploration leads to less contention between threads. Also note that these results were run prior to the rebuttal period, so do not reflect any of the changes described above. However, because the updated results in Table 3 of the attached pdf are not significantly different to the results from Table 1 of the paper, we don’t expect these results to significantly change when we re-run them with updated parameters.

Finally, since the submission, we have realised that the Alias method [1,2] for sampling from categorical can be used to significantly optimise action selection in all the algorithms we considered apart from PUCT. This improvement is an asymptotic improvement in complexity and we will add a complexity analysis in the appendix of the updated paper. In Table 5 in the attached pdf we give results on 19x19 Go using this optimisation, where a significant difference between the number of trials each algorithm was able to run can be seen between Tables 4 and 5. Again, because the updated results in Table 3 gave no substantial changes, we don’t expect any significant changes in these results either.

[1] Walker, Alastair J. "New fast method for generating discrete random numbers with arbitrary frequency distributions." Electronics Letters 8.10 (1974): 127-128.

[2] Vose, Michael D. "A linear algorithm for generating random numbers with a given distribution." IEEE Transactions on software engineering 17.9 (1991): 972-975.

---

> ### Author Response · Authors · 2023-08-21
> **Thank you to the reviewers**
>
> We just wanted to write a thank you to all four of our reviewer's, for the detailed reviews and insightful follow up questions, discussions and suggestions. It has been really helpful to get feedback on what needed more clarification and have a discussion with the reviewer's about how these things can be presented more clearly. Our experimental section has also been improved as a result of running additional experiments during this period. So again, thank you for the time taken to help us improve our work, we are very grateful

---

### Decision · Program_Chairs · 2023-09-21

**Decision:**

Accept (poster)

**Comment:**

The paper presents new Monte Carlo tree search variants based on using Boltzmann exploration. The paper not only proves several results about bounds for simple regret, it also shows several experiments on hard cases identified in the literature, standard benchmarks used in the literature (Sailing domain) and on 9x9 Go using the expert policy and value function from KataGo. Several reviewers had some criticisms including the motivations behind simple regret and questions about missing comparisons to baselines or mentions of previous work. The authors responded very well with several new experiments and clarifications. The discussions substantially affected the quality of the work and opinions of the reviewers. I agree with all of the reviewers' that much of the new content (and clarifications made in the discussion) should be incorporated in the camera-ready version of the paper, in the appendix if it does not fit in the main paper. Overall, this paper makes several significant novel contributions to the community.